# Investigation of Superconductivity in Ce-Doped (La,Pr)OBiS_2_ Single Crystals

**DOI:** 10.3390/ma15092977

**Published:** 2022-04-19

**Authors:** Masanori Nagao, Yuji Hanada, Akira Miura, Yuki Maruyama, Satoshi Watauchi, Yoshihiko Takano, Isao Tanaka

**Affiliations:** 1Center for Crystal Science and Technology, University of Yamanashi, 7-32 Miyamae, Kofu 400-0021, Japan; g20dte03@yamanashi.ac.jp (Y.H.); myuuki@yamanashi.ac.jp (Y.M.); watauchi@yamanashi.ac.jp (S.W.); itanaka@yamanashi.ac.jp (I.T.); 2International Center for Materials Nanoarchitectonics (MANA), National Institute for Materials Science, 1-2-1 Sengen, Tsukuba 305-0047, Japan; takano.yoshihiko@nims.go.jp; 3Faculty of Engineering, Hokkaido University, Kita-13 Nishi-8, Kita-ku, Sapporo 060-8628, Japan; amiura@eng.hokudai.ac.jp

**Keywords:** superconducting-related materials, single crystal growth, valence fluctuation

## Abstract

Single crystals of Ce-doped (La,Pr)OBiS_2_ superconductors, the multinary rare-earth elements substituted *R*OBiS_2_, were successfully grown. The grown crystals typically had a size of 1–2 mm and a plate-like shape with a well-developed *c*-plane. The c-axis lattice constants of the obtained (La,Ce,Pr)OBiS_2_ single crystals were approximately 13.6–13.7 Å, and the superconducting transition temperature was 1.23–2.18 K. Valence fluctuations of Ce and Pr were detected through X-ray absorption spectroscopy analysis. In contrast to (Ce,Pr)OBiS_2_ and (La,Ce)OBiS_2_, the superconducting transition temperature of (La,Ce,Pr)OBiS_2_ increased with the increasing concentrations of the tetravalent state at the *R*-site.

## 1. Introduction

Layered superconductors, such as cuprate [1,2,3] and iron-based superconductors [4,5], often exhibit high superconducting transition temperatures. Research on layered superconductors is important in order to understand materials that have high superconducting transition temperatures. *R*(O,F)BiS_2_ (*R*: rare-earth elements) compounds are layered superconductors and BiS_2_-based superconductors, and their superconductivity is triggered by substituting F at the O site [6,7,8,9,10,11]. A similar superconducting analogy is found in iron-based superconductors [5]. On the other hand, one of the BiS_2_-based superconductors, CeOBiS_2_, exhibits superconductivity without F substitution owing to the valence fluctuation between Ce^3+^ and Ce^4+^ [12,13]. Thus, F-free *R*OBiS_2_ compounds with partial Ce substitution at the *R*-site also become superconductors in the form of the (La,Ce)OBiS_2_ [14], (Ce,Pr)OBiS_2_ [15,16], (Ce,Nd)OBiS_2_ [17], and (La,Ce,Pr,Nd,Sm)OBiS_2_ [18] compounds. Moreover, *R*-site elements affect the superconducting transition temperature. Knowledge of the superconductivity of F-free *R*OBiS_2_ compounds is important for understanding BiS_2_-based superconductors. F-free *R*OBiS_2_ compounds with *R* = La, Ce, Pr have been obtained [12,13,19,20], but those of *R* = Nd, Sm have never been synthesized. Therefore, we focused on (La,Ce,Pr)OBiS_2_ single crystals.

In this paper, we successfully grew F-free (La,Ce,Pr)OBiS_2_ and (La,Pr)OBiS_2_ single crystals using a CsCl and a KCl flux, respectively. Hereafter, we denote binary and ternary *R*OBiS_2_ by the number of *R* elements: binary—(La,Ce)ObiS_2_, (Ce,Pr)ObiS_2_, and (La,Pr)OBiS_2_; ternary—(La,Ce,Pr)OBiS_2_. The obtained ternary single crystals of (La,Ce,Pr)OBiS_2_ were characterized by X-ray absorption fine structure (XAFS) spectroscopy for the Ce and Pr valence and the electrical transport properties down to approximately 0.2 K. The properties of the ternary (La,Ce,Pr)OBiS_2_ were investigated in comparison with those of binary *R*OBiS_2_ systems—namely, (La,Ce)OBiS_2_, (Ce,Pr)OBiS_2_, and (La,Pr)OBiS_2_. The relationship between the *R*-site valence state and the superconducting transition temperature (*T*_c_) for (La,Ce,Pr)OBiS_2_ was observed using the single crystals obtained.

## 2. Experimental

(La,Ce,Pr)OBiS_2_ and (La,Pr)OBiS_2_ single crystals were grown through a high-temperature flux method [12,20,21,22]. The raw materials—La_2_S_3_ (99.9 wt%), Ce_2_S_3_ (99.9 wt%), Pr_2_S_3_ (99.9 wt%), Bi_2_O_3_ (99.9 wt%), and Bi_2_S_3_ (99.9 wt%)—were weighed for a nominal composition of La*_a_*Ce*_b_*Pr*_c_*OBiS_2_ (*a* + *b* + *c* = 1.0). The mixture of the raw materials (0.8 g) and alkali metal chloride flux (5.0 g) was ground using a mortar and then sealed into an evacuated quartz tube (~10 Pa). The alkali metal chloride fluxes for the single crystal growths of (La,Ce,Pr)OBiS_2_ and (La,Pr)OBiS_2_ were CsCl (99.8 wt%) and KCl (99.5 wt%), respectively. The quartz tube was heated at *T*_max_ °C for 10 h, followed by a cooling to *T*_end_ °C at a rate of 1 °C/h. The *T*_max_ values for (La,Ce,Pr)OBiS_2_ and (La,Pr)OBiS_2_ were 950 °C and 1050 °C, respectively. The value of *T*_end_ depends on the melting temperature of the flux. In consequence, we adopted 650 °C and 750 °C for (La,Ce,Pr)OBiS_2_ and (La,Pr)OBiS_2_, respectively. The samples were then furnace-cooled to room temperature. The resulting quartz tube was opened in an air atmosphere, and the obtained products were washed and filtered by distilled water to remove the alkali metal chloride flux.

The compositional ratio of the single crystals was evaluated by energy-dispersive X-ray spectrometry (EDS) (Bruker; Quantax 70) associated with the observation of the microstructure using a scanning electron microscope (SEM) (Hitachi High-Technologies; TM3030). The obtained compositional values were normalized using La + Ce + Pr = 1.00, with the Bi and S values measured to a precision of two decimal places. The identification and evaluation of the orientation of the grown crystals were performed with X-ray diffraction (XRD) using Rigaku MultiFlex with Cu K*α* radiation. The superconducting transition temperature (*T*_c_) with zero resistivity was determined by the resistivity–temperature (*ρ*–*T*) characteristics. The *ρ*–*T* characteristics were measured using the standard four-probe method with a constant current density (*J*) mode and a physical property measurement system (Quantum Design; PPMS DynaCool). The electrical terminals were fabricated with Ag paste. The *ρ*–*T* characteristics in the temperature range 0.2–15 K were measured with an adiabatic demagnetization refrigerator (ADR) option for PPMS. The magnetic field for the operation of the ADR, 3 T at 1.9 K, was applied and subsequently removed. The temperature of the sample consequently decreased to approximately 0.2 K. The measurement of the *ρ*–*T* characteristics was begun at the lowest temperature (~0.2 K), which was spontaneously increased to 15 K. The valence state of the rare-earth elements (Ce, Pr) in the grown crystals was estimated using the X-ray absorption fine structure (XAFS) spectroscopy analysis with an Aichi XAS beamline and synchrotron X-ray radiation (BL5S1 and BL11S2). For the XAFS spectroscopy sample, the obtained single crystals were ground, mixed with boron nitride (BN) powder, and pressed into a pellet with a diameter of 4 or 10 mm.

## 3. Results and Discussion

Figure 1 shows a typical SEM image of the (La,Ce,Pr)OBiS_2_ single crystal. The obtained single crystals had plate-like shapes with sizes and thicknesses in the ranges of 1–2 mm and 100–400 μm, respectively. On the other hand, the obtained (La,Pr)OBiS_2_ single crystals exhibited plate-like shapes and were thin compared to the (La,Ce,Pr)OBiS_2_ single crystals. (La,Pr)OBiS_2_ single crystals became thick with increasing La contents. The ranges of their size and thickness were 0.5–1.0 mm and 10–200 μm, respectively.

Figure 2 shows the typical XRD patterns of a well-developed plane in the (La,Ce,Pr)OBiS_2_ and (La,Pr)OBiS_2_ single crystals obtained. The presence of only the 00*l* diffraction peaks, similar to CeOBiS_2_ compound structures [12], indicated a well-developed *c*-plane. The *c*-axis lattice constants of (La,Ce,Pr)OBiS_2_ and (La,Pr)OBiS_2_ single crystals were in the ranges of 13.59–13.68 Å and 13.80–13.81 Å, respectively. The differences between the *c*-axis lattice constants in the grown (La,Pr)OBiS_2_ single crystals were small, even though the compositions of the crystals varied extensively. Those values and the defined sample names are shown in Table 1. The analyzed atomic ratios of rare-earth elements in the grown (La,Ce,Pr)OBiS_2_ and (La,Pr)OBiS_2_ single crystals did not correspond precisely to the nominal compositions. The nominal compositions and the analyzed averaging compositions of the rare-earth elements are shown in Table 1. The estimated atomic ratios of the Bi and S elements in the obtained single crystals were Bi:S = 1.01 ± 0.05:2.01 ± 0.04, which agrees with the nearly stoichiometric ratio. On the other hand, Cs, K, and Cl from the flux were not detected in the single crystals with a minimum sensitivity limit of approximately 1 wt%.

Figure 3 shows the *ρ*–*T* characteristics parallel to the *c*-plane in the temperature range 0.2–15 K for the La_0.3__1_Ce_0.3__5_Pr_0.3__4_OBiS_2_ and La_0.6__0_Pr_0.__40_OBiS_2_ single crystals, which were typical (La,Ce,Pr)OBiS_2_ and (La,Pr)OBiS_2_ single crystals, respectively. La_0.3__1_Ce_0.3__5_Pr_0.3__4_OBiS_2_ single crystals exhibited superconductivity, but no superconducting transition was observed down to 0.2 K in La_0.6__0_Pr_0.__40_OBiS_2_ single crystals. The electrical resistivity slightly increased with the decreasing temperature, indicating a semiconducting behavior in the normal state. The resistivity of (La,Ce,Pr)OBiS_2_ single crystals in a normal state was far lower than that of (La,Pr)OBiS_2_. For this reason, we assumed that the carrier was induced because of the Ce valence fluctuation. The Ce valence state will be exhibited in Figure 5. The other obtained (La,Ce,Pr)OBiS_2_ and (La,Pr)OBiS_2_ single crystals also demonstrated similar behavior, except for the *T*_c_. Figure 4 shows the relationship between the superconducting transition temperature (*T*_c_) and the compositions of the rare-earth elements (La, Ce, Pr) analyzed for the binary and ternary *R*OBiS_2_ single crystals plotted on the ternary diagrams. The *T*_c_ values of (La,Ce)OBiS_2_ and (Ce,Pr)OBiS_2_ from previous reports are also shown in Figure 4 [12,13,14,15,16,19,20]. Superconductivity was not observed down to 0.2 K for the grown Ce-free *R*OBiS_2_ single crystals, which were (La,Pr)OBiS_2_ single crystals. On the other hand, the obtained (La,Ce,Pr)OBiS_2_ single crystals exhibited *T*_c_ values of 1.23–2.18 K. In the rare-earth element composition, *T*_c_ disappeared near the region with higher La contents. On the other hand, *T*_c_ was increased by increasing the Pr content. When the composition of the *R*-site became Ce_0.1_Pr_0.9_, *T*_c_ exhibited its maximum value. However, no superconductivity was observed in the end-component PrOBiS_2_ [20]. Furthermore, these results indicate that Ce substitution is required for superconductivity in binary and ternary *R*OBiS_2_. The CeOBiS_2_ superconductor was induced into superconductivity by a Ce valence fluctuation caused by a mixture state of trivalent (Ce^3+^) and tetravalent (Ce^4+^) electronic configurations [13]. Thus, we focused on the valence state of rare-earth elements in (La,Ce,Pr)OBiS_2_ single crystals.

Figure 5 shows (a) the Ce *L*_3_-edge and (b) the Pr *L*_3_-edge absorption spectra of the grown (La,Ce,Pr)OBiS_2_ single crystals and standard samples for each valence state using XAFS spectroscopy analysis at room temperature. The Ce *L*_3_-edge of the grown (La,Ce,Pr)OBiS_2_ single crystals demonstrated a peak at around 5725 eV and was assigned to trivalent electronic configuration (Ce^3+^) [23]. Moreover, the peaks around 5730 eV and 5737 eV were assigned to tetravalent electronic configuration (Ce^4+^) [24]. All grown (La,Ce,Pr)OBiS_2_ single crystals exhibited a Ce valence fluctuation caused by a mixture state of Ce^3+^ and Ce^4+^. The Ce valence states in the grown (La,Ce,Pr)OBiS_2_ single crystals were analyzed using linear combination fitting of Ce_2_S_3_ (Ce^3+^) and CeO_2_ (Ce^4+^) through XAFS spectroscopy spectra. In the tetravalent electronic configuration (Ce^4+^) peak, samples #2 and #3 demonstrated values around 5737 eV, which were low compared to those of other samples. The values of the Ce^4+^ concentrations at the *R*-site for these samples were close to those for samples #1 and #4, but the Ce element concentrations at the *R*-site were higher than those for samples #1 and #4 (See Table 2). In consequence, the Ce^4+^/Ce^3+^ ratios of samples #2 and #3 were smaller than those of samples #1 and #4. Therefore, the Ce^4+^ peaks became lower relative to those of samples #1 and #4. On the other hand, the Pr *L*_3_-edge of those single crystals exhibited a peak at approximately 5966 eV, which was assigned to the trivalent electronic configuration (Pr^3+^) [25,26]. Moreover, the tetravalent electronic configuration (Pr^4+^) demonstrated a peak at approximately 5978 eV in Pr_6_O_11_ [25,27]. The Pr valence states were also analyzed using a linear combination fitting of Pr_2_S_3_ (Pr^3+^) and Pr_6_O_11_ (one-third Pr^3+^ and two-thirds Pr^4+^). In consequence, the trivalent electronic structure was dominant, but a low concentration of the tetravalent electronic configuration was found. In a similar fashion, the Pr valence states in the grown (La,Pr)OBiS_2_ single crystals were analyzed. The tetravalent electronic structure (Pr^4+^) concentrations were low, as shown in Figure 6. The tetravalent electronic configuration (*R*^4+^) concentrations and the mean *R*-site ionic radius considering the valence state (*R*^3+^ and *R*^4+^) for the grown single crystals are summarized in Table 2. Exceptionally, the concentrations of Pr^4+^ in sample #1 (*R* = La_0.13_Ce_0.__20_Pr_0.6__7_) and sample #5 (*R* = La_0.__28_Pr_0.7__2_) were high, with both at approximately 10% at the *R*-site. The *T*_c_ of sample #1 exhibited comparatively high, but sample #5 did not exhibit superconductivity. Even though this observation still requires clarification, it provides important information on the superconducting mechanism of BiS_2_-based materials.

Herein, we discuss the effect of valence states and ionic radius on binary and ternary *R*OBiS_2_. Figure 7 shows the dependence of the tetravalent electronic configuration (*R*^4+^) concentrations on *T*_c_ for the binary and ternary *R*OBiS_2_ single crystals. While binary *R*OBiS_2_ exhibits little correlation between the ratio of tetravalent ions and *T*_c_, ternary *R*OBiS_2_ demonstrates a clear correlation between them; an increased ratio of tetravalent ions increased *T*_c_. This result indicates that the electronic state of ternary *R*OBiS_2_ is different from that of binary *R*OBiS_2_. Figure 8 shows the relationship between the mean *R*-site ionic radius [28] considering the valence state (*R*^3+^ and *R*^4+^) and the *T*_c_ for binary and ternary *R*OBiS_2_ single crystals. The *T*_c_ decreased with an increase in the mean *R*-site ionic radius. Although some anomalous data have been noted, this trend was consistent with the chemical pressure effect at the *R*-site [29]. There was no significant difference in terms of ionic radius between binary and ternary *R*OBiS_2_. Thus, ternary *R*OBiS_2_ is more sensitive to the electronic configuration, while both binary and ternary *R*OBiS_2_ have similar trends in their ionic radius. Additionally, these results reveal that Ce-substitution at the *R*-site is required for superconductivity.

## 4. Conclusions

(La,Ce,Pr)OBiS_2_ and (La,Pr)OBiS_2_ single crystals were successfully grown using a CsCl and a KCl flux, respectively. The superconducting transition temperature of the grown (La,Ce,Pr)OBiS_2_ single crystals was 1.23–2.18 K. (La,Pr)OBiS_2_ single crystals exhibited no superconductivity down to 0.2 K. The requirement of Ce substitution was revealed for the induction of superconductivity in binary and ternary *R*OBiS_2_. In the (La,Ce,Pr)OBiS_2_ and (La,Pr)OBiS_2_ single crystals, the Ce valence fluctuation was much larger than that of Pr, except for particular samples. For the ternary *R*OBiS_2_ (*R* = La + Ce + Pr) single crystals, the superconducting transition temperature increased with increasing concentrations of the tetravalent state (*R*^4+^) at the *R*-site, but the binary *R*OBiS_2_ (*R* = La + Ce, Ce + Pr, La + Pr) demonstrated no such correlation. According to the investigation of the mean *R*-site ionic radius, these superconducting transition temperature behaviors were found to be similar to the trend of the chemical pressure effect at the *R*-site. The superconducting transition temperature of *R*OBiS_2_ (*R* = La, Ce, Pr) did not demonstrate a simple dependence on the carrier concentration of the tetravalent state (*R*^4+^) at the *R*-site, but it may be easy to tune by varying the mean *R*-site ionic radius.

## Figures and Tables

**Figure 1 materials-15-02977-f001:**
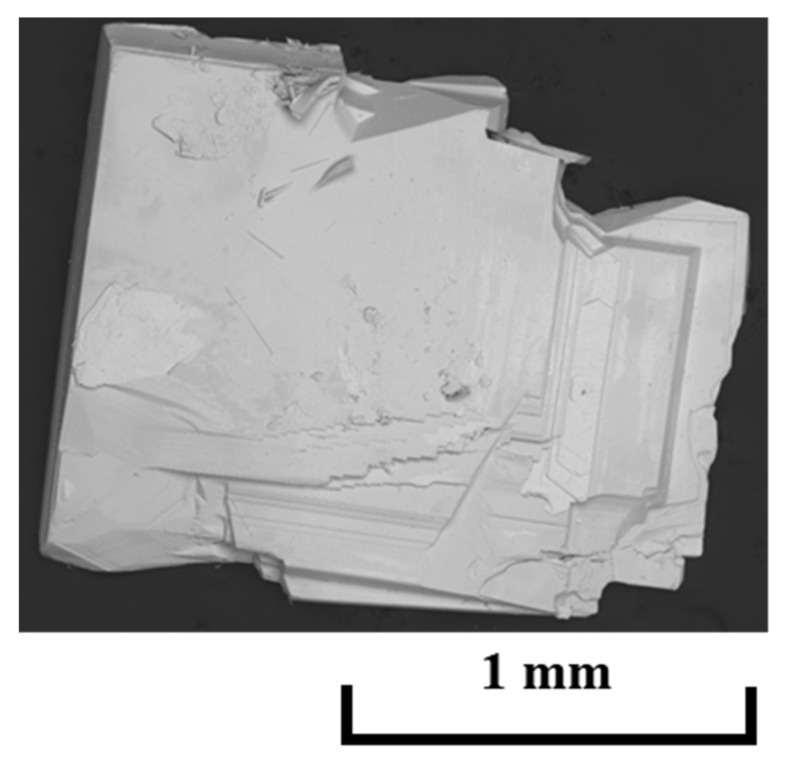
Typical SEM image of a (La,Ce,Pr)OBiS_2_ single crystal.

**Figure 2 materials-15-02977-f002:**
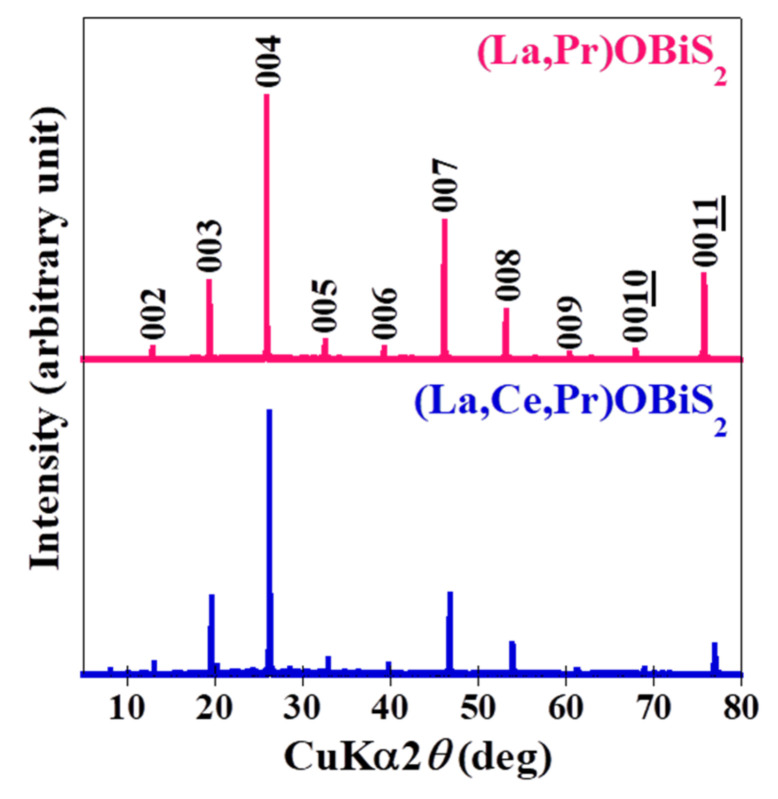
Typical XRD pattern of a well-developed plane of (La,CePr)OBiS_2_ and (La,Pr)OBiS_2_ single crystals.

**Figure 3 materials-15-02977-f003:**
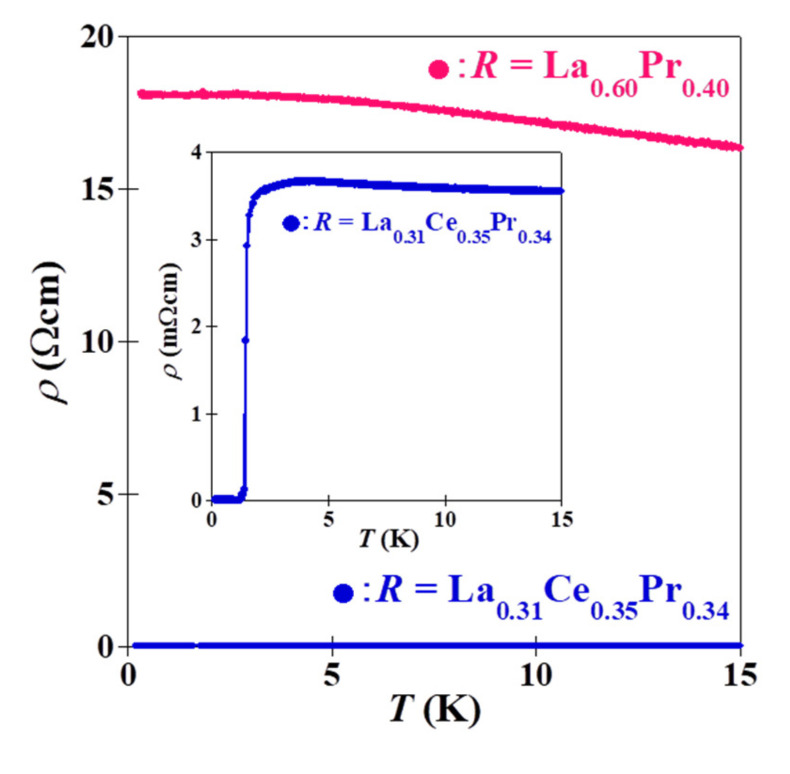
*ρ*–*T* characteristics parallel to the *c*-plane in the temperature range 0.2–15 K for the La_0.3__1_Ce_0.3__5_Pr_0.3__4_OBiS_2_ and La_0.6__0_Pr_0.__40_OBiS_2_ single crystals. The inset is an enlargement of the lower-resistivity region.

**Figure 4 materials-15-02977-f004:**
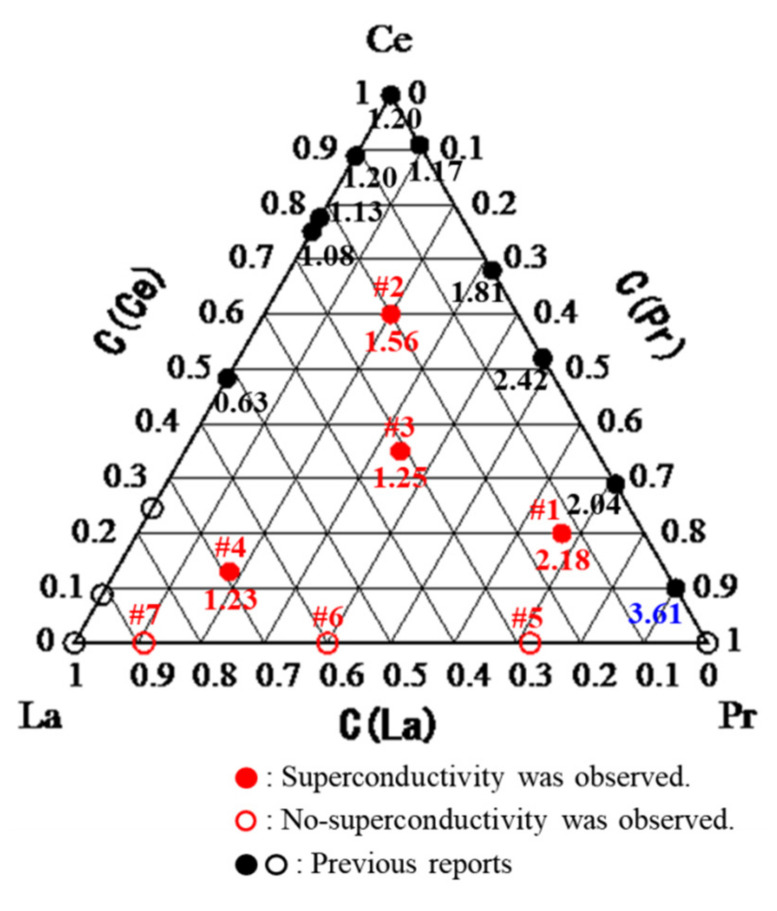
Relationship between the superconducting transition temperature (*T*_c_) and the compositions of the rare-earth elements (La,Ce,Pr) analyzed. The values in the ternary diagram are *T*_c_^zero^, in kelvin (K).

**Figure 5 materials-15-02977-f005:**
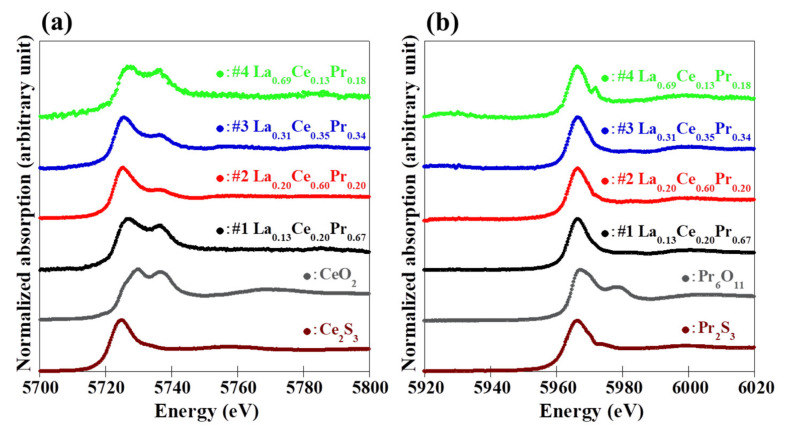
(**a**) Ce *L*_3_-edge and (**b**) Pr *L*_3_-edge absorption spectra using XAFS spectroscopy at room temperature for the grown (La,Ce,Pr)OBiS_2_ single crystals and standard samples for each valence state (Ce^3+^: Ce_2_S_3_; Ce^4+^: CeO_2_; and Pr^3+^: Pr_2_S_3_; Pr^4+^: Pr_6_O_11_).

**Figure 6 materials-15-02977-f006:**
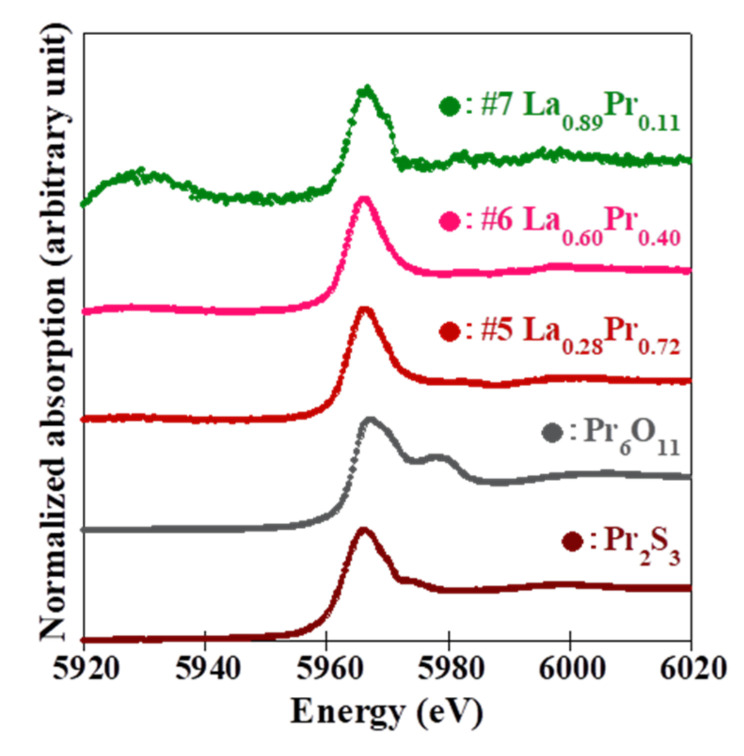
Pr *L*_3_-edge absorption spectra using XAFS spectroscopy at room temperature for the grown (La,Pr)OBiS_2_ single crystals, Pr_2_S_3_ and Pr_6_O_11_.

**Figure 7 materials-15-02977-f007:**
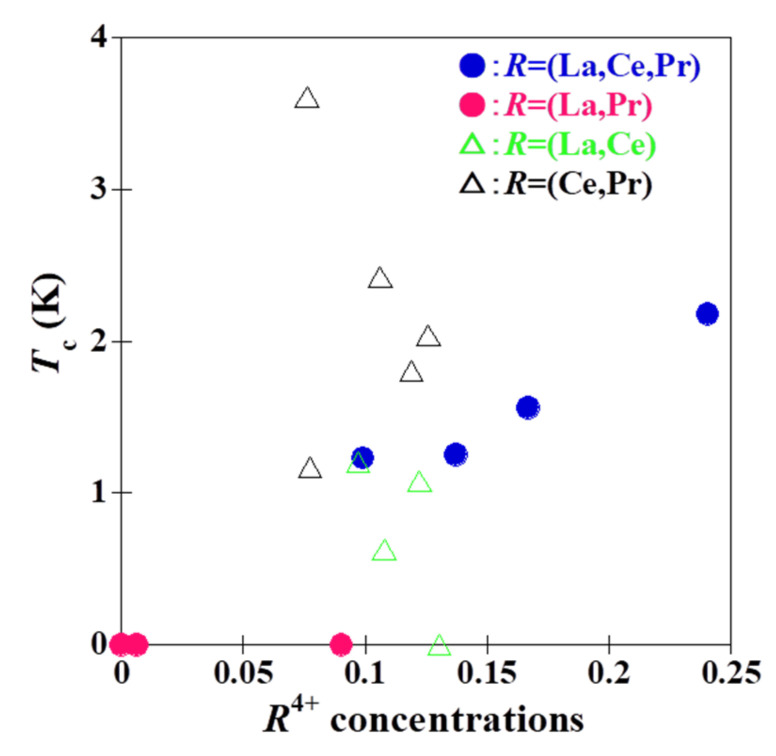
Dependence of the tetravalent electronic configuration (*R*^4+^) concentrations on *T*_c_ for the binary and ternary *R*OBiS_2_ single crystals.

**Figure 8 materials-15-02977-f008:**
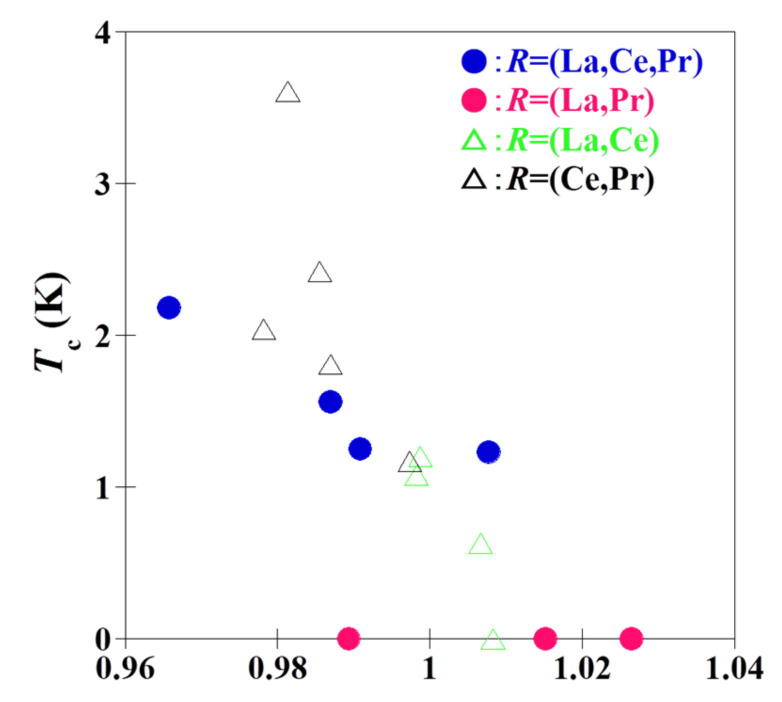
Relationship between the mean *R*-site ionic radius considering the valence state (*R*^3+^ and *R*^4+^) and the *T*_c_ for the binary and ternary *R*OBiS_2_ single crystals.

**Table 1 materials-15-02977-t001:** Nominal compositions and analyzed compositions at the *R*-site, and *c*-axis lattice constants in the grown single crystals.

Sample Name	Nominal Compositions			Analyzed Compositions			*c*-Axis Lattice Constants (Å)
	La:*a*	Ce:*b*	Pr:*c*	La	Ce	Pr	
#1	0.10	0.20	0.70	0.13 ± 0.01	0.20 ± 0.01	0.67 ± 0.02	13.68
#2	0.25	0.50	0.25	0.20 ± 0.02	0.60 ± 0.03	0.20 ± 0.02	13.59
#3	0.33	0.33	0.33	0.31 ± 0.01	0.35 ± 0.04	0.34 ± 0.05	13.61
#4	0.65	0.15	0.20	0.69 ± 0.03	0.13 ± 0.01	0.18 ± 0.04	13.68
#5	0.10	0	0.90	0.28 ± 0.02	0	0.72 ± 0.03	13.80
#6	0.50	0	0.50	0.60 ± 0.02	0	0.40 ± 0.01	13.80
#7	0.90	0	0.10	0.89 ± 0.03	0	0.11 ± 0.03	13.81

**Table 2 materials-15-02977-t002:** Tetravalent electronic configuration (*R*^4+^) concentrations at the *R*-site, and mean *R*-site ionic radius considering the valence state (*R*^3+^ and *R*^4+^) for the grown single crystals.

Sample Name	Analyzed Averaging Compositions			Tetravalent Electronic Configuration (*R*^4+^) Concentrations at the *R*-Site			Mean *R*-Site Ionic Radius (Å)
	La	Ce	Pr	Ce^4+^	Pr^4+^	Total (Ce^4+^+Pr^4+^)	
#1	0.13	0.20	0.67	0.13	0.11	0.24	0.966
#2	0.20	0.60	0.20	0.16	0.01	0.17	0.987
#3	0.31	0.35	0.34	0.11	0.03	0.14	0.991
#4	0.69	0.13	0.18	0.09	0.01	0.10	1.01
#5	0.28	0	0.72	0	0.09	0.09	0.989
#6	0.60	0	0.40	0	0	0	1.02
#7	0.89	0	0.11	0	0.01	0.01	1.03

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
