# Peer review of "Investigation of Superconductivity in Ce-Doped (La,Pr)OBiS2 Single Crystals"

_materials, 2022, doi:10.3390/ma15092977_

Round 1

Reviewer 1 Report

The MS is well written and equipped with new results. The same can be published as is. 

Author Response

Thank you very much for your positive comments. 

Reviewer 2 Report

Based on my experience, this is a report, not a paper.

Results are not novel and their quality are not considerable.

Authors selected a material and did some experiments so that they presented an industrial report with low quality in Figs and justifications.

Author Response

We think these data are important for understanding BiS2-based superconductors, and other reviewers replied with positive responses. We have revised this manuscript, and then we leave a decision up to the editor. 

Reviewer 3 Report

The work done by Nagao et al has reported the superconductivity of ROBiS2 compounds, particularly focusing on the effect of the rare earth element Ce. Their study suggests Ce substitution is important for superconductivity because of the valence fluctuation and also chemical pressure. The work is carefully done and indeed provides more information for the superconductivity study of ROBiS2. I recommend this manuscript be published in MDPI Materials if the authors can respond to my comments appropriately.

  1. The resistivity of LaPrOBiS2 and LaCePrOBiS2 in figure 3 are very different. Is there a reason? Have you measured the transport along the perpendicular direction? 
  2. The c lattice constant in the table1 is interesting. The c of sample 3 is the smallest, which is surprisingly corresponding to the equal number of La, Ce, Pr i.e., La0.33Ce0.33Pr0.33. Is this sample superconducting?
  3. The authors used XAS to determine the element valence. Do these valence assignments still give a charge neutral system?
  4. The author emphasizes the importance of the Ce valence fluctuation and the chemical pressure. How about the low-energy electronic structure details, such as the Fermi surface? Because Ce can be 4+, I expect the Ce substitution will shift the Fermi level. Is there any result from angular resolved photoemission spectroscopy?
  1. The presentation needs to be improved since some sentence is confusing.

Author Response

>The resistivity of LaPrOBiS2 and LaCePrOBiS2 in figure 3 are very different. Is there >a reason?

Response

Ce substitution extremely decreases the normal state resistivity in ROBiS2 single crystals. We assumed that the carrier is induced due to the Ce valence fluctuation.

We added the following sentence:

The resistivity of (La,Ce,Pr)OBiS2 single crystals in a normal state was far lower than that of (La,Pr)OBiS2. For this reason, we assumed that the carrier was induced because of the Ce valence fluctuation. The Ce valence state is exhibited in Figure 5.

> Have you measured the transport along the perpendicular direction?

We have never measured the transport property along the perpendicular direction.

>The c lattice constant in the table1 is interesting. The c of sample 3 is the smallest, >which is surprisingly corresponding to the equal number of La, Ce, Pr i.e., >La0.33Ce0.33Pr0.33. Is this sample superconducting?

Response

The smallest c-axis lattice constant is sample #2 which becomes superconductivity. Sample #3 with an equal number of La, Ce, Pr elements also exhibited superconductivity.

>The authors used XAS to determine the element valence. Do these valence >assignments still give a charge neutral system?

Response

We were not able to clarify the charge-neutral in the obtained single crystals, because the concentration of oxygen with the anion role cannot be measured for high accuracy.

>The author emphasizes the importance of the Ce valence fluctuation and the chemical >pressure. How about the low-energy electronic structure details, such as the Fermi >surface? Because Ce can be 4+, I expect the Ce substitution will shift the Fermi level. >Is there any result from angular resolved photoemission spectroscopy?

Response

We have never been performed the angular resolved photoemission spectroscopy.

>The presentation needs to be improved since some sentence is confusing.

Response

We have revised some sentences.

Reviewer 4 Report

The manuscript by M. Nagao et al., deals with the single crystal growth and observation of the superconducting transition of (La,Pr,Ce)OBiS2. The samples are analyzed with SEM+EDS, XAS, resistivity and XRD. While the physics behind are interesting and feature new insight that value publication, the current state of the text, the quality of figures and the overall presentation is not acceptable. In many cases the meaning cannot be understood although the meant statement might be intersting, e.g.: “According to the investigation of the mean R-site ionic radius, those superconducting transition temperatures behaviors were observed similarly to the trend of the chemical pressure effect in the R-site.”

For the EDS results an error bar obtained by statistical distribution comparing several points would be helpful as the precision of EDS is not very high.

The manuscript has a large amount of typos/grammatical errors as:

Page 1 The sentence misses were mixed with: “Alkali metal chloride fluxes for (La,Ce,Pr)OBiS2 and (La,Pr)OBiS2 were CsCl (99.8 wt%) and KCl (99.5 wt%), respectively.”

Page 2 Remove the before air: “The resulting quartz tube was opened in the air,”

Page 5 For instead of to: “single crystals showed similar behavior except to the”

Page 5 the what?: “Figure 4 summarized the between”

Page 5 “No superconductivity at down to 0.2 K was observed”

....

The manuscript requires a detailed rewriting before being considered for publication.

Author Response

>For the EDS results an error bar obtained by statistical distribution comparing several >points would be helpful as the precision of EDS is not very high.

Response

We added the range of error for the analyzed compositions in Table I.

>The manuscript has a large amount of typos/grammatical errors as:

>Page 1 The sentence misses were mixed with: “Alkali metal chloride fluxes for >(La,Ce,Pr)OBiS2 and (La,Pr)OBiS2 were CsCl (99.8 wt%) and KCl (99.5 wt%), >respectively.”

>Page 2 Remove the before air: “The resulting quartz tube was opened in the air,”

>Page 5 For instead of to: “single crystals showed similar behavior except to the”

>Page 5 the what?: “Figure 4 summarized the between”

>Page 5 “No superconductivity at down to 0.2 K was observed”

>The manuscript requires a detailed rewriting before being considered for publication.

Response

We have revised some sentences and performed the English language editing service.

Reviewer 5 Report

The manuscript prepared by Nagao et al. studies the superconductivity of Ce-doped (La,Pr)OBiS2 single crystals. I am overall positive regarding this work. Thus, I suggest the publication after minor revision. Authors should address few comments below:

  1. More details could be added to the introduction section. For example, why superconductors are desired and what practical applications they could be used for. What materials in history have been studied as superconductors? Why R(O,F)BiS2 compounds are more desirable compared to the other studied superconductor materials in history (i.e., what are the benefits of R(O,F)BiS2 compounds)? Please cite more references to give readers a better introduction for the motivation this research.
  2. Authors should raise the quality of few figures. For example, subscripts for La, Ce, and Pr elements in fig. 2 are hard to tell.
  3. Authors may consider cutting the x-axis of XRD plot (fig. 2) starting from 5 deg to remove the left white space.

Author Response

>More details could be added to the introduction section. For example, why >superconductors are desired and what practical applications they could be used for. >What materials in history have been studied as superconductors? Why R(O,F)BiS2 >compounds are more desirable compared to the other studied superconductor >materials in history (i.e., what are the benefits of R(O,F)BiS2 compounds)? Please cite >more references to give readers a better introduction for the motivation this research.

Response

We have revised and added some sentences in the introduction section.

>Authors should raise the quality of few figures. For example, subscripts for La, Ce, >and Pr elements in fig. 2 are hard to tell.

Response

We have replaced these Figures.

>Authors may consider cutting the x-axis of XRD plot (fig. 2) starting from 5 deg to >remove the left white space.

Response

We have executed it. 

Reviewer 6 Report

This works deals with single crystal superconductors that are great of importance in the field. The manuscript provides good results from detailed analyses. The paper is well-structured and clearly give the main idea. I suggest publication after some revisions. Here my observations.

  • Introduction is quite weak to give the state-of-the-art. I suggest discussing previous works instead of giving them in a bulk citing.
  • Related to the previous comment, after giving the current technology in the field, authors should emphasize the contribution of their work. They should clearly state the novelty in this study.
  • Please give the application areas for these single crystals. In which sectors and for what purposes are they used?
  • Most of the figures have low resolutions. Please increase the image quality.
  • In Fig.5a, the second peak around 5730 is quite humble for #2 while it is more visible for the others. Why?
  • I think Conclusions should be revised by giving the main concluding remarks and discussing the benefits in applications that can be achieved by these results.

Author Response

>Introduction is quite weak to give the state-of-the-art. I suggest discussing previous >works instead of giving them in a bulk citing.

>Related to the previous comment, after giving the current technology in the field, >authors should emphasize the contribution of their work. They should clearly state the >novelty in this study.

>Please give the application areas for these single crystals. In which sectors and for >what purposes are they used?

Response

We have revised and added some sentences in the introduction section.

>Most of the figures have low resolutions. Please increase the image quality.

Response

We have replaced these Figures.

>In Fig.5a, the second peak around 5730 is quite humble for #2 while it is more visible >for the others. Why?

Response

We think this reason is due to the Ce concentration in the R-site.

We added the following sentence:

In the tetravalent electronic configuration (Ce4+) peak, samples #2 and #3 showed values around 5737 eV, which were low compared to those of other samples. The values of the Ce4+ concentrations at the R-site for these samples were close to those for samples #1 and #4, but the Ce element concentrations at the R-site were higher than those for samples #1 and #4 (See Table II). In consequence, the Ce4+/Ce3+ ratios of samples #2 and #3 were smaller than those of samples #1 and #4. Therefore, the Ce4+ peaks became lower relative to those of samples #1 and #4.

>I think Conclusions should be revised by giving the main concluding remarks and >discussing the benefits in applications that can be achieved by these results.

Response

We have added the sentence in the conclusions section. 

Round 2

Reviewer 2 Report

Accept